# Process Optimization of Electrochemical Oxidation of Ammonia to Nitrogen for Actual Dyeing Wastewater Treatment

**DOI:** 10.3390/ijerph16162931

**Published:** 2019-08-15

**Authors:** Jiachao Yao, Yu Mei, Guanghua Xia, Yin Lu, Dongmei Xu, Nabo Sun, Jiade Wang, Jun Chen

**Affiliations:** 1College of Biology and Environmental Engineering, Zhejiang Shuren University, Hangzhou 310015, China; 2College of Environment, Zhejiang University of Technology, Hangzhou 310014, China

**Keywords:** electro-oxidation, ammonia, N_2_ selectivity, process optimization, kinetics

## Abstract

To mitigate the potential environmental risks caused by nitrogen compounds from industrial wastewater, residual ammonia after conventional wastewater treatment should be further eliminated. In this work, an electrochemical oxidation process for converting ammonia to nitrogen in actual dyeing wastewater was investigated. The effects of the main operating parameters, including initial pH value, applied current density, NaCl concentration, and flow, were investigated on ammonia removal and products distribution. Experimental results indicated that, under optimal conditions of an initial pH value of 8.3, applied current density of 20 mA cm^−2^, NaCl concentration of 1.0 g L^−1^, and flow of 300 mL min^−1^, the ammonia could be completely removed with N_2_ selectivity of 88.3% in 60 min electrolysis. A kinetics investigation using a pseudo-first-order model provided a precise description of ammonia removal during the electro-oxidation process. Experimental functions for describing the relationships between kinetic constants of ammonia removal and main operating parameters were also discussed. Additionally, the mechanisms and economic evaluation of ammonia oxidation were conducted. All these results clearly proved that this electro-oxidation process could efficiently remove ammonia and achieve high N_2_ selectivity.

## 1. Introduction

Ammonia is a major toxic pollutant in the discharged water of industries that leads to the eutrophication of ecosystems, which is fatal to aquatic lives, and is a hindrance to the disinfection of water supplies [1,2,3]. Stricter discharge standard for ammonia in the effluent of industrial wastewater treatment is required in China. The biological method is one of the most effective and economical processes for ammonia-containing wastewater treatment [4,5]. However, continuous monitoring (i.e., pH, temperature, and carbon source) is needed for microorganism growth [6]. Several other methods, such as ion exchange [7,8] and reverse osmosis [9], are also used to remove ammonia. However, concentrated wastewater will be produced during the process which may pose a disposal problem [10,11].

Recently, electrochemical oxidation has attracted lots of attention and has been considered as a promising alternative for wastewater treatment [12,13]. In the recent literature, many scholars have investigated the oxidation performance of ammonia in simulated wastewater [14,15]. For example, Shu et al. [16] reported that, for simulated heavy metal wastewater treatment, ammonia could be removed from 80 mg L^−1^ to 0.1 mg L^−1^ in 2.5 h electrolysis at 30 mA cm^−2^. In addition, many researches focusing on the oxidation behavior of ammonia over modified electrodes have been reported [17,18]. Li et al. [19] reported the ammonia removal kinetics and performance with an SnO_2_–CNT filter anode and a Pd−Cu/NF cathode, indicating that rapid and effective transformation of ammonia to nitrogen could be achieved. Shih et al. [20] investigated the electrochemical ammonia oxidation over a nickel foam-supported Ni(OH)_2_/NiOOH electrode, suggesting that 98.8% ammonia could be removed at current density of 1.5 mA cm^−2^ in 7 h. In general, many studies have demonstrated that ammonia can be eliminated and converted to N_2_ by electrochemical oxidation using thermally decomposed iridium oxide film, boron-doped diamond, Ti/IrO_2_, Ti/RuO_2_, etc. [21,22].

However, there are few studies on electrolytic decomposition characteristics and kinetics for ammonia removal in actual wastewater [23,24]. Additionally, the side reaction of ammonia over-oxidation, i.e., the generation of nitrate and nitrite, always occurs during the whole electrochemical process [25,26], especially in the treatment of actual wastewater because of its characteristics of complexity and variety [27]. Thus, it seems that achieving high ammonia removal efficiency and improving the product selectivity of N_2_ are one of the prerequisites for industrial application of electrochemical wastewater treatment. In order to relieve these issues, it is essential to find suitable operating parameters to achieve the oxidation performance of ammonia to N_2_. As reported by Zou et al. [28] and Chen et al. [29], an electrochemical method for wastewater treatment is complicated, and strongly affected by electrode material, applied current density, NaCl concentration, pH, flow rate, temperature, etc. It means that the ability of electrochemical conversion of ammonia to N_2_ can be highly improved by controlling the main operating parameters.

In this study, the process optimization of electrochemical oxidation of ammonia to nitrogen for actual wastewater treatment was investigated with a Ti/PbO_2_ anode and a Ti cathode. The effects of the main operating parameters, including initial pH value, applied current density, NaCl concentration, and flow were studied. The results were evaluated by considering the ammonia removal efficiency and products distribution. A tentative description of the kinetics of ammonia removal was also presented. Besides, the mechanisms and economic feasibility of ammonia oxidation were discussed.

## 2. Materials and Methods

### 2.1. Wastewater Characteristics and Electrode Selection

The actual wastewater was collected from the secondary sedimentation tank of the wastewater treatment process at a dyeing factory located in the Shaoxing region of Zhejiang Province, China. The characteristics of the wastewater are shown in Table 1.

Electrode material plays a key role in the electro-oxidation process. Commercial anodes of Ti/IrO_2_, Ti/RuO_2_, and Ti/PbO_2_ have been compared in our previous study [30]. According to the oxygen evolution potential, removal efficiency of pollutants, and electrode life, Ti/PbO_2_ shows an excellent ability on oxidation of ammonia and organics. Another aspect of our work [31] has shown that Ti/PbO_2_ has high catalytic activity for oxidizing ammonia in simulated wastewater via direct and indirect oxidation. Thus, Ti/PbO_2_ is selected as the experimental anode in this study. Ti is selected as the cathode in this work because of its practicability, high stability, and low cost.

### 2.2. Electrolysis System

The electrochemical oxidation system for ammonia removal mainly consisted of a 0.4 L reaction cell, a water bath, DC power supply, reservoir, and gas collection device, as shown in our previous work [31]. The reaction cell was a single-compartment electrochemical cell containing a Ti/PbO_2_ anode and Ti cathode (Baoji Special Metals Co., Ltd., Baoji, China). The anode and cathode were both square (10 × 10 cm^2^) with mesh-like structures. The inter-electrode gap was 20 mm. The water bath was used to maintain the temperature of the electrolysis solution at the desired set point (30 °C).

### 2.3. Analytical Methods

During experiments, samples were drawn from the reactor at certain intervals and then analyzed for the concentrations of ammonia, nitrate, nitrite, nitrogen, and so on. Ammonia was analyzed by Nessler reagent spectrophotometry. Nitrate (NO_3_^−^), nitrite (NO_2_^−^), and chloride (Cl^−^) ion concentrations were measured by ion chromatography (ICS-1100, Dionex, Sunnyvale, CA, USA). Nitrogen oxides and nitrogen gas were monitored by gas chromatography (GC-6890, Agilent, Santa Clara, CA, USA). The concentration of the hydroxyl radical was studied by the N, N-dimethyl-p-nitrosoaniline (RNO) method [29]. RNO was used for spin trapping the hydroxyl radicals, and the bleaching of the yellow color (RNO) during the electrolysis process was also measured at 440 nm using a spectrophotometer. Active chlorine (i.e., Cl_2_, HClO, and ClO^−^) was measured following the DPD standard method using the N, N-diethyl-p-phenylenediamine reagent [12].

The removal efficiency *R* (%) of ammonia was determined using Equation (1):(1)$$R=\frac{C_{0}-C_{t}}{C_{0}}\times 100\%$$
where *C*_0_ is the initial ammonia concentration (mg L^−1^) and *C_t_* is the ammonia concentration at electrolysis time *t*.

The selectivity (*S*, %) of N_2_ was calculated as follows:(2)$$S\;=\;\frac{C_{N}}{C_{0}-C_{t}}\;\times \;100\%$$
where *C_N_* is the produced concentration of N_2_ (mg-N L^−1^).

The related pseudo-first-order kinetic model is given by [32]:(3)$$\ln\frac{C_{t}}{C_{0}}\;=\;-Kt$$
where *K* is apparent reaction rate constant of pseudo-first-order kinetics (min^−1^) and *t* is the reaction time (min).

The current efficiency *CE* (%) and energy consumption (kWh g^−1^) were calculated based on the following equations [22]:(4)$$CE=\frac{3(C_{0}-C_{t})FV}{14It},$$
(5)$$E=\frac{UIt}{V(C_{0}-C_{t})},$$
which *F* is the Faraday number (96,485 C mol^−1^); *V* is the volume of the wastewater (L); *I* is the current (A); *U* is the voltage (*V*); *3* is the number of exchanged electrons for electro-oxidation of ammonia; and 14 is the atomic mass of N (g mol^−1^). All experimental data were averaged from three independent experiments, and the errors were calculated to be less than 8%.

## 3. Results

### 3.1. Influence of Initial pH Value

Solution pH is one of the important factors that affects the performance of the electrochemical process. As a consequence, the effect of initial pH values ranging from 2.7 to 12.0 on ammonia removal was investigated. Diluted H_2_SO_4_ and NaOH solutions were used for pH adjustments, and a pH of 8.3 was the natural pH of the dyeing wastewater.

Figure 1 presents the ammonia removal and products distribution in function of initial pH values. As shown in Figure 1a, increasing the initial pH value had a great influence on the ammonia removal; that is, the removal efficiencies were 58.9%, 70.9%, 41.9%, 77.5%, and 80.6% for initial pH values of 2.7, 4.8, 6.9, 8.3, and 12.0, respectively. The results indicated that the highest electrocatalytic performance for ammonia oxidation was obtained in alkaline conditions, while that with a neutral condition was the lowest. Additionally, Figure 1b displays that ammonia removal followed pseudo-first-order kinetics [33], and the apparent reaction rate constants (K) are listed in Table 2. The value of K increased about two times as the initial pH rose from 2.7 to 12.0, however it got the lowest at pH 6.9. This phenomenon might be associated with the existence of forms of active chlorine in different pH values. To be specific, Cl_2(aq)_ and HClO are the predominant species in acidic conditions, whereas HClO and ClO^−^ mainly exist in alkaline condition [28]. The generated Cl_2(aq)_ might escape into the atmosphere, leading to the reduction of active chlorine which could oxidize ammonia in acidic conditions. Besides, our previous work had determined that ammonia could be removed via direct electron transfer on Ti/PbO_2_ in a wide pH range, but is weakest in neutral conditions [31].

Figure 1c shows the concentration profiles of N_2_, NO_3_^−^, and NO_2_^−^ during ammonia oxidation after 80 min electrolysis. In acidic conditions, N_2_ was the main product of ammonia degradation, NO_3_^−^ was less than 0.6 mg-N, and no NO_2_^−^ or NOx (including NO and N_2_O) was detected. The N_2_ selectivities of 98.7% and 99.7% were achieved at initial pH values of 2.7 and 4.8, respectively. In alkaline conditions, non-negligible amounts of NO_3_^−^ were produced, especially at an initial pH of 12.0; approximately 93.2% and 69.4% of removed ammonia were convert to N_2_ at pH 8.3 and 12.0, respectively, suggesting that the higher the initial pH value was, the easier the accumulation of NO_3_^−^/NO_2_^−^ would occur. This phenomenon might be related to the cathodic reduction of NO_3_^−^/NO_2_^−^, as shown in Equations (6) and (7) [34], i.e., strong alkaline conditions were unsuitable for reduction of the generated NO_3_^−^/NO_2_^−^ on the cathodes.
(6)$$NO_{2}{}^{-}+2H_{2}O+3e^{-}\to 1/2N_{2}+4OH^{-},$$
(7)$$NO_{3}{}^{-}+3H_{2}O+5e^{-}\to 1/2N_{2}+6OH^{-}.$$

According to the ammonia removal and products distribution, an initial pH value of 8.3 was considered as the optimal one in this case, i.e., the pH did not need to be adjusted before electrochemical treatment.

### 3.2. Influence of Applied Current Density

Applied current density, which is defined as electric current per unit cross-section area of the electrodes, is a key experimental parameter that affects the electro-oxidation process as it regulates the capability of active radical generation on the electrode surface. Figure 2 shows the function of applied current density on ammonia removal during the electro-oxidation of dyeing wastewater.

As shown in Figure 2a, the ammonia removal increased significantly with increase of applied current density, and the removal efficiencies were 47.9%, 63.2%, 72.4%, 77.5%, and 85.3%, respectively. This phenomenon could be ascribed to the fact that the increase in applied current density could accelerate the forming of active radicals, and then increase the ammonia oxidation rate. Figure 2b indicates that ammonia removal in different current densities followed pseudo-first-order kinetics, and the apparent reaction rate constants are listed in Table 2. Additionally, the relationship between the apparent reaction rate constant K and the applied current density J is described in Figure 2c, and the exponential function for the relationship is shown in Equation (8):(8)K=−0.030exp(−0.035J)+0.033.

Substituting Equation (8) into the pseudo-first-order kinetic model (Equation (3)), and integrating it gives the equations for the reaction kinetics:(9)$$\ln(C_{t}/C_{0})=[0.030\exp(-0.035J)-0.033]\cdot t.$$

Equation (9) is the pseudo-first-order kinetic model developed to describe the evolution of ammonia during the electrochemical treatment at different applied current densities.

Figure 2d shows the products distribution of ammonia oxidation after 80 min electrolysis. The N_2_ selectivities increased from 47.1% to 93.2% as the applied current densities increased from 5 to 20 mA cm^−2^; then it decreased slightly when further increasing applied current density. This result might be related to cathodic reduction; that is, though NO_3_^−^/NO_2_^−^ was generated during ammonia oxidation on the anode (Equations (10)−(12)) [25], the NO_3_^−^/NO_2_^−^ could be efficiently removed on the cathode (Equations (6) and (7)) as applied current densities increased from 5 to 20 mA cm^−2^; then a further increase in applied current density accelerated the hydrogen evolution reaction and competed with the NO_3_^−^/NO_2_^−^ reduction on cathode [34].
(10)$$NH_{3}-6e^{-}+2H_{2}O\to NO_{2}{}^{-}+7H^{+},$$
(11)$$NH_{3}-8e^{-}+3H_{2}O\to NO_{3}{}^{-}+9H^{+},$$
(12)$$NH_{4}{}^{+}+4HClO\to NO_{3}{}^{-}+H_{2}O+6H^{+}+4Cl^{-}.$$

Taking into account the above results, the applied current density of 20 mA cm^−2^ would be favorable for ammonia removal.

### 3.3. Effect of Added NaCl Concentration

Active chlorine is one of the most common electrochemically generated oxidants for indirect oxidation of ammonia [35]. In order to obtain the exact influence of indirect oxidation mediated by active chlorine, the effect of added NaCl concentration was investigated on ammonia electro-oxidation, as shown in Figure 3.

Figure 3a shows the variation of ammonia removal during electrolysis with different added NaCl concentrations. It was observed that the removal efficiency of ammonia increased obviously from 77.5% to 95.8% with addition of 0.5 g L^−1^ NaCl; with further increases in the NaCl concentration, the removal efficiency reached 100%. The results suggested that increasing NaCl concentration promoted the generation of active chlorine which was beneficial to the ammonia indirect oxidation. Besides, Figure 3b displays that the pseudo-first-order model could precisely describe the kinetics of ammonia removal. The values of rate constant K, corresponding to the different NaCl concentrations, are shown in Table 2 and Figure 3c. It could be seen that there was a polynomial relationship between the obtained K and NaCl concentration, as expressed in Equation (13):(13)$$K=-1.39\times 10^{-3}{\left[\mathrm{NaCl}\right]}^{2}+1.90\times 10^{-2}[\mathrm{NaCl}]+1.33\times 10^{-3}.$$

Integrating the equation subject to Equation (3), we obtain:(14)$$\ln(C_{t}/C_{0})=(1.39\times 10^{-3}{\left[\mathrm{NaCl}\right]}^{2}-1.90\times 10^{-2}[\mathrm{NaCl}]-1.33\times 10^{-3})\cdot t.$$

Equation (14) describes the kinetics of ammonia removal along the electrochemical wastewater treatment as a function of the NaCl concentration in the range of 0–1.5 g L^−1^.

Figure 3d is the result of products distribution during ammonia oxidation with different NaCl concentrations. The N_2_ selectivities were 93.3%, 88.4%, 87.6%, and 84.8% for the NaCl concentration of 0, 0.5, 1.0, and 1.5 g L^−1^, respectively. The results indicated that an increase in NaCl concentration could aggravate the over-oxidation of ammonia, i.e., increase the NO_3_^−^/NO_2_^−^ generation, which was consistent with the findings by Mandal et al. [36].

Considering the ammonia removal and N_2_ selectivity, the added NaCl concentration of 1.0 g L^−1^ might be the optimized value.

### 3.4. Impact of Flow

The flow is an important parameter to determine the mechanistic mass transfer rate and the retention time in the reactor. In generally, the increase of flow can improve the mass-transport of the pollutants to the electrode surface where they react with electro-generated radicals, and consequently increase the efficiency of the process.

Figure 4 shows the ammonia removal with respect to the four flows (50, 100, 200, and 300 mL min^−1^). As shown in Figure 4a, the flow had a positive effect on ammonia removal: The removal efficiency increased from 92.0% to 100% as flow increased from 50 to 100 mL min^−1^ in 80 min electrolysis; with a further increase in flow, ammonia could be completely removed in a shorter electrolysis time. Meanwhile, Figure 4b displays that the removal of ammonia had an exponential relationship with electrolysis time, and the slopes (i.e., rate constant K) were 3.12 × 10^−2^, 4.385 × 10^−2^, 5.865 × 10^−2^, and 6.547 × 10^−2^ for 50, 100, 200, and 300 mL min^−1^, respectively. As shown in Figure 4c, the obtained K value presented a polynomial relationship with flow, as expressed in Equation (15).
(15)$$K=-4.842\times 10^{-7}F^{2}+3.045\times 10^{-4}F+1.755\times 10^{-2}.$$

Substituting Equation (15) into Equation (3), the finial reaction kinetic equation was gotten:(16)$$\ln(C_{t}/C_{0})=(4.842\times 10^{-7}F^{2}-3.045\times 10^{-4}F-1.755\times 10^{-2})\cdot t.$$

Equation (16) could be used to express the kinetics of ammonia removal with different flows.

Figure 4d illustrates the concentration profiles of NO_3_^−^ and N_2_ during the electrochemical treatment of dyeing wastewater at flows in the range of 50–300 mL min^−1^. The results indicated that, though flow could affect the ammonia removal efficiency, there was little effect on the N_2_ selectivity. The selectivity of N_2_ was stable at 87.0%–88.3%.

Considering the ammonia removal and electrolysis time, a flow of 300 mL min^−1^ was thus selected as the optimized one.

### 3.5. Mechanisms and Economic Evaluation

Mechanisms of electrochemical oxidation of ammonia are divided into direct and indirect anodic oxidation. The detail mechanism of direct oxidation on the Ti/PbO_2_ electrode has been fully described in our previous research [37]. Briefly, in direct electrolysis, pollutants are oxidized after transferred to the anode surface, without involvement of any substances other than the electron, which is a clean reagent (Equation (17)) [38]. Oxidizing agents such as the hydroxyl radical and active chlorine are added in the case of indirect anodic oxidation to react with ammonia. As shown in Figure 5, both the hydroxyl radical and active chlorine increased with electrolysis time, suggesting that ammonia could be efficiently oxidized by these oxidants, as shown in Equations (18) and (19). Besides, no chloramine was detected. The COD and chromaticity in wastewater were completely removed within 20 min.
(17)$$2NH_{3}-6e^{-}+6OH^{-}\to N_{2}+6H_{2}O,$$
(18)$$2NH_{3}+6\cdot OH\to N_{2}+6H_{2}O,$$
(19)$$2NH_{4}{}^{+}+3HClO\to N_{2}+3H_{2}O+5H^{+}+3Cl^{-}.$$

The technical feasibility of the electro-oxidation is usually evaluated in terms of pollutant removal efficiency, while the economic feasibility is determined by the current efficiency and energy consumption. Figure 6 shows the ammonia removal efficiency with electrolysis time and the corresponding current efficiency, under the optimum conditions. The results indicated that the variation of ammonia removal with electrolysis time was opposite to that of current efficiency. After 60 min electrolysis, the removal efficiency of 100% was achieved, suggesting that this electro-oxidation process was efficient enough to meet the strict requirements of the Discharge Standards of Water pollutants for Dyeing and Finishing of Textile Industry in China (ammonia < 20 mg-N L^−1^). Meanwhile, an N_2_ selectivity of 88.3% was achieved. Additionally, current efficiency of 18.5% was obtained, which was slightly higher than that of other studies using an electro-oxidation process for ammonia removal. For instance, Chen et al. [39] reported a current efficiency for ammonia oxidation of 10% at a current density of 5 mA cm^−2^ with a RuO_2_–IrO_2_–TiO_2_/Ti anode. Kim et al. [40] studied the electrocatalytic oxidation of ammonia to nitrogen in a self-made reaction cell, and a current efficiency of 13% was achieved with the operating conditions of 80 mA cm^−2^ and a pH of 13. Furthermore, the energy consumption for the treatment of per milligram ammonia was calculated as 0.16 kWh g^−1^ in this work.

## 4. Conclusions

A process optimization of electrochemical oxidation of ammonia to nitrogen for actual dyeing wastewater treatment was studied. The initial pH value, applied current density, NaCl concentration, and flow were investigated as the main operating parameters on ammonia removal. At an optimized initial pH value of 8.3, applied current density of 20 mA cm^−2^, NaCl concentration of 1.0 g L^−1^, and flow of 300 mL min^−1^, the electro-oxidation process could completely remove ammonia in 60 min with a Ti/PbO_2_ anode and a Ti cathode. Kinetics investigations using pseudo-first-order kinetics provided a suitable description of ammonia removal. The apparent reaction rate constants for ammonia removal were obtained as functions of the applied current density, NaCl concentration, and flow, respectively. All the results indicated that the optimized electro-oxidation process could efficiently convert ammonia to nitrogen in actual dyeing wastewater treatment.

## Figures and Tables

**Figure 1 ijerph-16-02931-f001:**
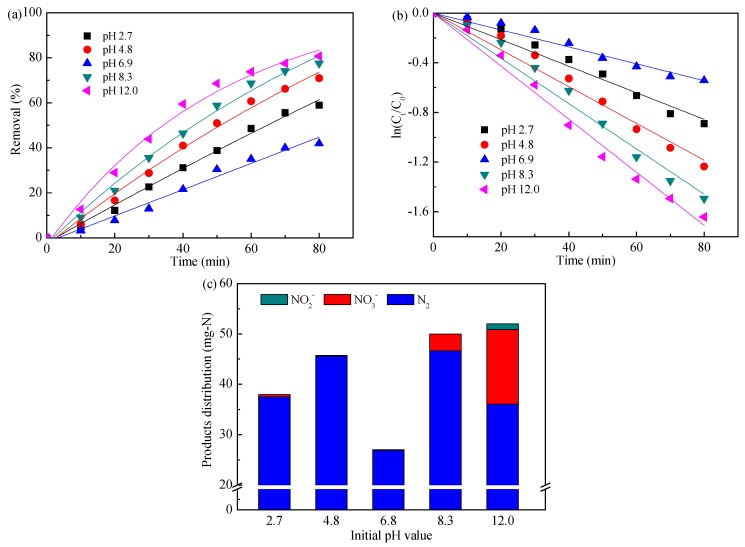
Influence of initial pH value on ammonia removal (**a**), kinetics analysis (**b**), and products distribution (**c**). Applied current density of 20 mA cm^−2^ and flow of 100 mL min^−1^. Diluted H_2_SO_4_ and NaOH solutions were used for pH adjustments.

**Figure 2 ijerph-16-02931-f002:**
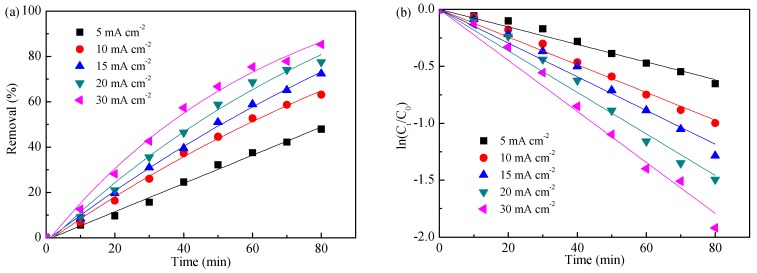
Function of applied current density on ammonia removal (**a**), kinetics analysis (**b**), kinetics relationship (**c**), and products distribution (**d**). Initial pH value of 8.3 and a flow of 100 mL min^−1^.

**Figure 3 ijerph-16-02931-f003:**
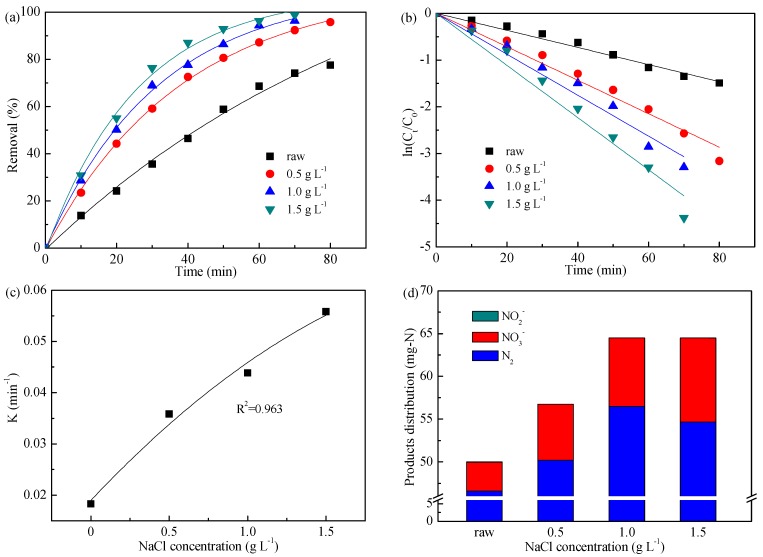
Effect of NaCl concentration on ammonia removal (**a**), kinetics analysis (**b**), kinetics relationship (**c**), and products distribution (**d**). Initial pH value of 8.3, applied current density of 20 mA cm^−2^, and a flow of 100 mL min^−1^.

**Figure 4 ijerph-16-02931-f004:**
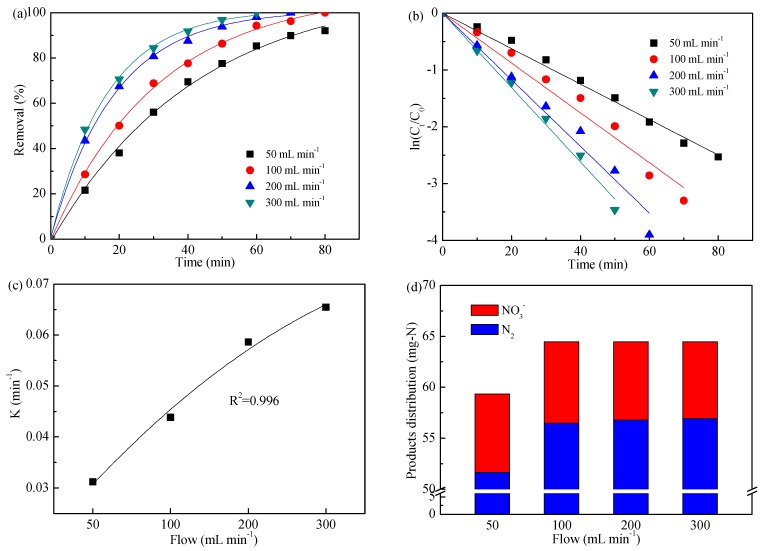
Impact of flow on ammonia removal (**a**), kinetics analysis (**b**), kinetics relationship (**c**), and products distribution (**d**). Initial pH value of 8.3, applied current density of 20 mA cm^−2^, and NaCl concentration of 1.0 g L^−1^.

**Figure 5 ijerph-16-02931-f005:**
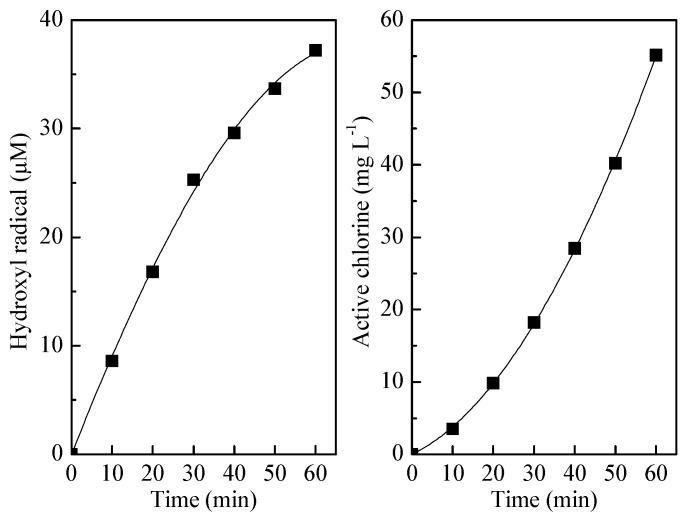
Generation of the hydroxyl radical and active chlorine. Initial pH value of 8.3, applied current density of 20 mA cm^−2^, NaCl concentration of 1.0 g L^−1^, and a flow of 300 mL min^−1^.

**Figure 6 ijerph-16-02931-f006:**
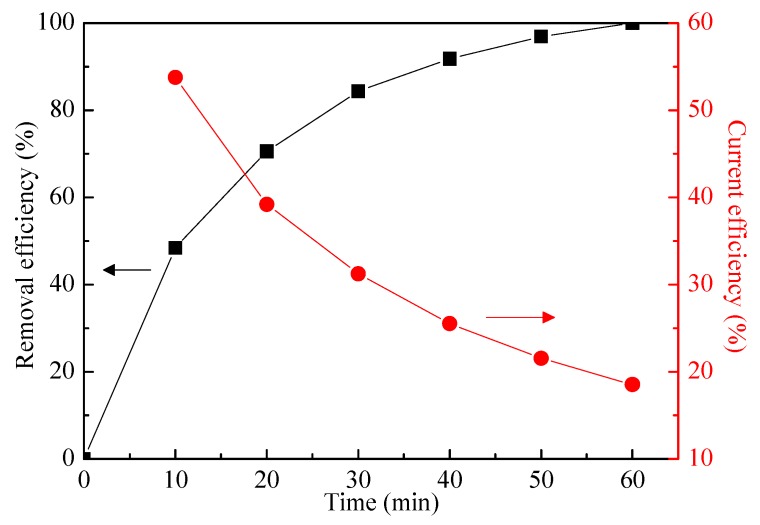
The ammonia removal and current efficiency during the electro-oxidation process. Initial pH value of 8.3, applied current density of 20 mA cm^−2^, NaCl concentration of 1.0 g L^−1^, and a flow of 300 mL min^−1^.

**Table 1 ijerph-16-02931-t001:** The physicochemical characteristics of actual dyeing wastewater used in this study.

Parameters	Units	Actual Wastewater
pH	-	8.3 ± 0.1
Ammonia	mg-N L^−1^	161.2 ± 2.0
Chemical oxygen demand (COD)	mg L^−1^	58.8 ± 3.5
Chloride (Cl^−^)	mg L^−1^	483.3 ± 1.3
Sodium (Na^+^)	mg L^−1^	1343.3 ± 2.6
Sulfate (SO_4_^2^^−^)	mg L^−1^	1221.6 ± 1.5
Chromaticity	times	25
Conductivity	mS cm^−1^	6.61 ± 0.1

**Table 2 ijerph-16-02931-t002:** The apparent reaction rate constants of ammonia removal with different initial pH values, applied current densities, NaCl concentrations, and flows.

Parameters	Conditions	100 × K (min^−1^)	R^2^
Initial pH value	2.7	1.069	0.988
	4.8	1.480	0.989
	6.9	0.680	0.986
	8.3	1.822	0.990
	12.0	2.135	0.996
Applied current density (mA cm^−2^)	5	0.770	0.992
	10	1.216	0.995
	15	1.479	0.991
	20	1.822	0.990
	30	2.239	0.993
NaCl concentration (g L^−1^)	0	1.822	0.990
	0.5	3.583	0.991
	1.0	4.385	0.988
	1.5	5.582	0.988
Flow (mL min^−1^)	50	3.120	0.996
	100	4.385	0.988
	200	5.863	0.990
	300	6.547	0.996

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
