# Peer review of "Process Optimization of Electrochemical Oxidation of Ammonia to Nitrogen for Actual Dyeing Wastewater Treatment"

_ijerph, 2019, doi:10.3390/ijerph16162931_

Round 1
Reviewer 1 Report
The primary contribution of this manuscript by Yao et al. is to optimize electrochemical oxidation of ammonia to nitrogen for actual dyeing wastewater treatment by using Ti/PbO2 anode. The authors investigated the effect of initial pH, current density, NaCl concentration and flow velocity on ammonia removal, thereby they could obtain the optimized ammonia removal approaching 100% and 88.3% N2 selectivity. Overall, the review finds the manuscript to be of novel insight that may be a good contribution to engineered application of electrochemical oxidation process. However, the authors may want to consider the following suggestions to make sure that all the concerns can be clarified before it can be published in International Journal of Environmental Research and Public Health.
(1) the authors should justify the reason why they selected Ti/PbO2 electrode for electrochemical oxidation of ammonia. This question is closely related to the mechanism of ammonia removal, but remains to be unclearly discussed by the authors. PbO2 is the typical anode having high potential of oxygen evolution reaction to produce OH from water oxidation. But if the mechanism of ammonia oxidation is mediated by HClO rather than OH radicals, it seems unnecessary to use PbO2 anode for two reasons below. (i) the potential required for Cl2 evolution is much lower than that for OER, making it possible apply a wide range of electrode materials except for PbO2. (ii) there is potential risk of dissolution of PbO2 at high potential, leading to secondary pollution of Pb2+ released to water.
(2) Since the authors carried out their study by using one-compartment electrolytic cell, it is necessary to elucidate the interaction between anode and cathode, such as pH variation and its impact to ammonia removal.
(3) the authors need to provide more information on the quality of actual wastewater, particularly the organic content such as COD or TOC. Although the study focused on ammonia removal, the data of organic abatement is also important.
(4) The methods for quantification of OH should be described briefly rather than it is provided by only reference 25.
(5) When investigating the impact of flow velocity, the mass transfer coefficient can be discussed on the basis of dimensionless correlation of Sh=f(Re, Sc).
(6) The authors have to double check with the logics and English writing throughout the manuscript.
Reviewer 2 Report
The manuscript entitled “Process optimization of electrochemical oxidation of ammonia to nitrogen for actual dyeing wastewater treatment” studies the ammonia removal and products distribution, looking for the selectivity to form nitrogen by electrochemical oxidation with Ti/PbO2 anode. The influence of initial pH value, current density, NaCl concentration and flow has been studied. This is a very interesting work with good results, only some aspects must be completed. Thus, a minor revision is recommended for this manuscript.
Specifically, these points must be improved:
- The term “flow velocity” is not suitable, it is better to use only “flow”.
- Introduction:
· Line 45: Why direct oxidation? Ammonia can be also removed by indirect oxidation by the action of the electrogenerated oxidizing agents…
· Line 46: The anodic material used in the references 20 and 21 should be specified.
- Materials and Methods:
· Table 1: Taking into account that the analysed effluent is from the secondary treatment of a wastewater treatment plant, the values of ammonia and chloride concentration are very high, why?
· Line 83: What is the set point for the temperature?
· Line 90: What is “active chlorine”? ClO-? •Cl? This must be specified and revised in all the paper.
· Line 101: Why pseudo-first order kinetic? Is the hydroxyl radicals and other oxidants concentration within the kinetic constant? This must be explained and justified with references:
A.M. Polcaro, S. Palmas, F. Renoldi, M. Mascia, On the performance of Ti/SnO2 and Ti/PbO2 anodes in electrochemical degradation of 2-chlorophenol for wastewater treatment, J. Appl. Electrochem. 29 (1999) 147–151.
Martín de Vidales, M.J.; Robles-Molina, J; Dominguez-Romero, J.C.; Cañizares, P.; Sáez, C.; Molina-Díaz, A.; Rodrigo M.A. Removal of sulfamethoxazole from waters and wastewaters by conductive-diamond electrochemical oxidation. J. Chem. Technol. Biotechnol. 87, 2012, 1441-1449.
· Line 106: 14 is not the molecular mass of ammonia, it is the atomic mass of N. This must be corrected.
- Results and discussion:
· Line 113: H2SO4 is used to maintain the pH at the desired set point, however, you are adding •SO4 for the action of the electrochemical oxidation in the acid. Thus, it is better to use HCl. In this context, efficiency increases with the pH, and this can be due to a recombination of oxidants (for an excessive generation of them), that can take place when H2SO4 is added (for low values of pH). Therefore, this must be considered and explained in the text.
These references can be used to explain this behaviour:
Martín de Vidales, M.J.; Barba, S.; Sáez, C.; Cañizares, P.; Rodrigo, M.A. Coupling ultraviolet light and ultrasound irradiation with Conductive-Diamond Electrochemical Oxidation for the removal of progesterone. Electrochim. Acta 140, 2014, 27-32.
Martín de Vidales, M.J.; Sáez, C.; Pérez, J.F.; Cotillas, S.; Llanos, J.; Cañizares, P.; Rodrigo M.A. Irradiation-assisted electrochemical processes for the removal of persistent organic pollutants from wastewater. J. Appl. Electrochem. 45, 2015, 799-808.
· Figure 1a: Removal efficiency cannot be evaluated versus reaction time. Thus the y-axis must be modified to “Removal (%)”. In addition, the maximal value of this axis must be 100. These considerations must be also taken into account in Figures 2, 3 and 4.
· Figure 1b: pH values of 4.8 and 6.8 do not entail the formation of NO2- and NO3-. Thus, are not these values better because the formation of more toxic species is not occurring? This must be taken into account in the next figures.
· Line 152: The title must be “Influence of the applied current density”.
· Lines 157-164: When the applied current density is not constant, the process efficiency must be evaluated regarding the applied electric charge (A·h·dm-3). The representation versus reaction time is only to evaluate the degradation rate.
· Lines 171-177: Reactions that are taking place (on the anode and the cathode) should be shown.
· Point 3.3: Is the formation of ClO3- and ClO4- possible? This must be considered in the different experiments, because these species present a high toxicity in the medium.
· Line 221: The word “pollutions” must be changed to “pollutants”.
· Line 252: The pollutants are not adsorbed on the anodic surface. This is not true, the electrochemical oxidation is a surface process.
· Line 257: Were chloramines measured? How? What was the measured concentration?

Reviewer 3 Report
The submitted paper “Process optimization of electrochemical oxidation of ammonia to nitrogen for actual dyeing wastewater treatment” aimed to remove ammonia in secondary effluent from a dyeing factory. The best performance was obtained and removal pathway of ammonia was discussed. This study may be helpful to some degree from the viewpoint of practical engineering aspect to enhance removal. However, it is obvious that the paper lacked in novelty with comparison of other published works. There are already lots of research focused on the electrooxidation on ammonia. The author need to illustrate the difference and advantage of this paper in this field. Thus, the manuscript still needs greater revision before the consideration of further publication. More detailed comments are listed in the following:
1. Introduction- more illustration and reference on electrooxidation processes for ammonia removal should be added in the introduction. Please highlight the novelty of this paper.
2. Materials and Methods-The characteristic of secondary effluent is doubtful, where initial ammonia and conductivity was too high. Please provide the content of other anions, such as sulfate, nitrate…..
Materials and Methods-how to detect Nitrogen gas?
3. Fig.1- Eq(5) and (6) is hard to occur in undivided system. The reason of nitrogen conversion section need more reference and data to support. In other words, the description on transformation between ammonia, nitrite, and nitrate in section 3.1-3.4 was unclear. The interpretation of data need to be analyzed in-depth.
4. Fig.3a- please explain the obvious enhancement of ammonia removal from 77.5% to 95.8% when chloride ion increased from 483.3mg/L to 500 mg/L.
5. Fig.5- The content of hydroxyl radical and active chlorine in different conditions is of importance for the illustration of ammonia removal mechanisms. Thus, the comparison of oxidants under different chloride concentration is essential to be added.
6. For the consideration of practical engineering, the variation of COD and color also need to be investigated under the role of hydroxyl radical and active chlorine in situ generated.
7. Section 3.5- Economic evaluation could not be found here. Energy cost of this electrochemical process should be provided to ensure this technology is a promising application for dye wastewater treatment.
Round 2
Reviewer 3 Report
The manuscript has been improved on writing. Many mistake and unnecessary information had been revised and deleted. The advantage of the process was illustrated by authors more clearly this time, although the novelty of this paper is still limited. Some suitable publications were listed, and the highlights were refined. The entire paper has been much improve.